# Effect of Calcium Carbide Residue on Strength Development along with Mechanisms of Cement-Stabilized Dredged Sludge

**DOI:** 10.3390/ma15134453

**Published:** 2022-06-24

**Authors:** Xinyi Zhu, Fujun Niu, Lu Ren, Chenglong Jiao, Haiqiang Jiang, Xiaoyue Yao

**Affiliations:** 1State Key Laboratory of Subtropical Building Science, School of Civil Engineering and Transportation, South China University of Technology, Guangzhou 510641, China; 202021009185@mail.scut.edu.cn (X.Z.); 202020107833@mail.scut.edu.cn (L.R.); ctcljiao@mail.scut.edu.cn (C.J.); 202021009193@mail.scut.edu.cn (X.Y.); 2South China Institution of Geotechnical Engineering, School of Civil Engineering and Transportation, South China University of Technology, Guangzhou 510641, China; 3School of Water Conservancy and Civil Engineering, Northeast Agricultural University, Harbin 150030, China; hqiangjiang@126.com

**Keywords:** calcium carbide residue, dredged sludge, strength development, microstructure, soil improvement

## Abstract

The purpose of this research is to explore the feasibility of using calcium carbide residue (CCR), a by-product from acetylene gas production, as a solid alkaline activator on the strength development in CCR–Portland cement-stabilized dredged sludge (CPDS). The effects of cement content, CCR content and curing time on the strength development of CPDS were investigated using a series of unconfined compressive strength (UCS), pH and electric conductivity (EC) tests. Scanning electron microscopy and X-ray diffraction analyses were performed to gain additional insight into the mechanism of strength development. Meanwhile, the carbon footprints of CPDS were calculated. Following the results, it was found that CCR can significantly improve the strength of cemented dredged sludge. On the basis of the strength difference (ΔUCS) and strength growth rate (UCS_gr_), it was recommended that utilizing 20% cement with the addition of 20% CCR is the most effective way to develop the long-term strength of CPDS. In addition, the microstructural analysis verified that the optimum proportion of CCR benefits the formation of hydration products in CPDS, particularly needle-like gel ettringite, resulting in a less-porous and dense inter-locked structure. Furthermore, the solidification mechanism of CPDS was discussed and revealed. Finally, it was confirmed that CCR can be a sustainable alternative and effective green alkaline activator for the aim of improving cemented dredged sludge.

## 1. Introduction

With the development of engineering constructions, river dredging, lake dredging and the construction of port channels result in the inevitable large production of dredged sludge [1,2,3], which causes both environmental and ecological damage. Traditional disposal methods such as dumping offshore or burying in the garbage dump would cause unavoidable environmental and ecological hazards to the surrounding environment. Dredged sludge has the natural characteristics of high compressibility, high water content, low bearing capacity and rich organic matter [4,5,6]. That is, the untreated dredged sludge is difficult to be used directly in engineering due to its poor engineering properties. In this context, soil solidification is one of the effective methods for dealing with the gradually increasing dredged sludge [7]. Through a series of chemical reactions, the cured dredged sludge not only solves the environmental problems caused by stacked sludge but also can be transformed into construction materials (e.g., road embankments, structural backfill, etc.).

Portland cement (PC) is the most common curing agent and is widely used for soft soil improvement and solidification of dredged sludge, since it is readily available with a significant curing effect at a low price [8,9,10,11,12,13]. The main curing mechanism of cemented soil is through the hydration, which results in the formation of a variety of hydration products, including calcium silicate hydrates (CSH) and calcium aluminate hydrates (CAH), thus increasing the strength [14,15,16,17,18]. However, the application of cement as a stabilizer has raised serious concerns due to its environmental issues:(i) quarries need to be expanded for cement production; (ii) significant CO_2_ emissions (approximately 0.95 ton) from manufacturing and other associated industrial processes, which contributes 5–7% of global carbon dioxide emissions; (iii) the production of non-beneficial by-products, which can cause cement–soil to usually exhibit high alkalinity, is detrimental to groundwater quality [19,20,21,22]. Thus, based on the aim of reducing the consumption of cement while improving the strength of solidified soil, the addition of small amounts of the activator may be a promising approach.

Typical activators include quicklime, hydroxides of alkali metals or alkaline earth metals, carbonates, sulfates and silicates. Among them, the most commonly used materials are quicklime, sodium silicate and sodium hydroxide. Quicklime has long been used as a curing agent for the solidification of soft soil because of the remarkable curing effect [19,20]. It is usually used as an activator agent together with cement for curing soft soils [23,24]. However, the production of quicklime, same as cement, consumes large amounts of natural resources, leading to negative environmental impacts. In addition, the preparation of sodium silicate and sodium hydroxide also consumes significant amounts of electrical energy [25]. These traditional alkaline activators are not only expensive but also wasteful on natural resources in the generation process, and they even are poisonous in some cases. Due to the disadvantage of traditional alkaline activators, it is crucial to find a substitute for them.

Calcium carbide residue (CCR) is a by-product of the acetylene industry, with calcium hydroxide as the primary component. According to statistics, there are many acetylene gas production devices all over the world, resulting in a large amount of CCR production. Among them, China is the largest manufacturer and consumer of calcium carbide around the world, accounting for 90–95% of global supply and demand, and the annual output of dry CCR in China is 900,000–1,140,000 t [26]. In recent years, CCR has been widely utilized for cement manufacturing [26,27,28], but its total utilization rate remains still low (no more than 10%) [29]. Moreover, most of them are landfilled in waste dumps, which not only results in valuable land occupation but also causes huge contamination to the surrounding environment including the pollution of ground water and soil due to its high alkalinity [30]. Because CCR is chemically and mineralogically similar to hydrated lime, CCR has the potential to replace lime in civil engineering [31]. Previous studies proved that soft clays can be successfully treated with CCR. Horpibulsuk et al. [32] investigated the use of CCR as a binder for the treatment of over-wet clay soils used to fill embankment materials and showed that CCR-stabilized soils outperform lime-stabilized soils. Latifi et al. [33] used CCR to improve mechanical properties of green bentonite, demonstrating the potential of CCR as a sustainable alternative to conventional stabilizers. Moreover, the effectiveness of CCR has also been studied in conjunction with other environmentally friendly industrial by-products, including blast furnace refining slag [34] and biomass ash [35]. For example, Yi et al. [34] used CCR to excite GGBS for the improvement of soft soils and found that the mechanical properties of CCR-GGBS stabilized soils outperform those of Portland cement stabilized soils.

So far, CCR has been used itself alone or in conjunction with other industrial by-products as a soil stabilizer. However, there is very limited information on CCR used as a solid alkaline activator to substitute traditional activators for cement binder. The application of CCR as an eco-friendly activator in cement-stabilized dredged sludge is thus novel and crucial among the geotechnical fraternity. Therefore, the aim of this study is to investigate the effect of CCR as a solid alkaline activator, an environmentally friendly alternative to traditional activators, to improve mechanical properties of cement-stabilized dredged sludge. A variety of tests including unconfined compressive strength (UCS), pH and electric conductivity (EC) were conducted to investigate the physical–chemical and mechanical properties of CCR-Portland cement-stabilized dredged sludge (CPDS). An array of microstructure tests were conducted to fully reveal the possible curing mechanisms. Furthermore, the carbon footprints of CPDS were calculated and compared with those of cemented dredged sludge. Finally, the solidification mechanism of CPDS was discussed and revealed in this research. The outcome of this study would utilize large amounts of CCR from landfill sites and reduce carbon emissions from Portland cement production significantly.

## 2. Materials and Methods

### 2.1. Materials

The dredged sludge was collected in a construction site in Dawang High-tech Zone, Zhaoqing City, Guangdong Province, China (Figure 1). The sampling was performed in test pits, 2.0 m in length by 1.2 m in width and 8.5 m in depth, underlain by a dark gray clay with rotten trunks. According to the American Society of Testing and Material (ASTM), the basic physical characteristics were detected, and the properties of dredged sludge are listed in Table 1. The particle size distribution curve of dredged sludge is shown in Figure 2, and the soil can be classified as high plasticity clay (CH) based on the Unified Soil Classification System.

Two additives used are Portland cement (P.O.42.5) and calcium carbide residue (CCR). The Portland cement was produced by Nanjing Conch Cement Co., Ltd., and the CCR powder was obtained from a waste calcium carbide residue dump in Guangzhou, China. The chemical components determined by X-ray fluorescence (Axios, PANalytical Co., Amsterdam, The Netherlands) according to the China Building Materials Test and Certification Center (CBMCC) (2009) are listed in Table 2, showing that the main constituents are identified as SiO_2_, CaO, and Fe_2_O_3_.

Furthermore, the microstructure of dredged sludge obtained from SEM (MIRA LMS, TESCAN Co., Prague, Czechia) analysis is provided in Figure 3, indicating that the dredged sludge particles are irregularly shaped and incompact, which is made up of several microscopic polymers with a lamellar structure. The crystalline phase of dredged sludge revealed by XRD (Ultima IV, Rigaku Co., Tokyo, Japan) measurements is shown in Figure 4, demonstrating that the dredged sludge is primarily composed of quartz and calcite.

### 2.2. Mixture Design and Experimental Procedure

The dry jet mixing method is often used in engineering construction for the solidification of dredged sludge with high water content. Hence, the experiment was carried out by simulating this method, and the mixture design of stabilized dredged sludge is described in Table 3. The water content of stabilized dredged sludge was set at 60%, which is the mass ratio to the dry dredged sludge. The content of Portland cement binder is the mass ratio to dry dredged sludge, and the content of CCR activator is the mass ratio to the binder. Furthermore, different curing times of 3 days, 7 days and 28 days were selected.

The preparation process of the samples is shown in Figure 5. The original dredged sludge was first dried in an oven at 105 °C for 24 h and then crushed into powder, which passed through the 2 mm sieves. On the basis of experimental design, the requisite amounts of water, dry dredged sludge, and additives were determined and weighed. Afterward, the weighed dredged sludge and additives were mechanically blended in the mixer for 3 min until uniformly distributed; then, they were mixed with water and kept mixing for another 3 min to ensure uniformity. The well-mixed CPDS was then poured in three layers into molds (39.1 mm in diameter and 80 mm in height), and both ends of the molds were sealed with two plastic films. For the sake of making specimens homogeneous and removing the gas brought by the mixing process, the molds were placed on a shaking table for 50–60 s after pouring each layer of specimens. Then, the prepared CPDS were sent to the standard curing room (20 ± 2 °C, 90 ± 5% RH) for 24 h. Finally, after curing with molds, the samples were removed and sealed with plastic containers and continually sent to a standard curing room until the design curing time.

### 2.3. Testing Methods

#### 2.3.1. Unconfined Compressive Strength Test

The unconfined compressive strength (PPS, HUAKAN TECHNALOGY Co., Beijing, China) was tested for all the samples after 3 days, 7 days and 28 days of curing, according to ASTM [36]. The experiments were performed with a constant displacement of 2 mm/min. Three parallel tests were taken to ensure the accuracy for each stabilized soil.

#### 2.3.2. pH and Electrical Conductivity Tests

The pH and EC tests were performed on CPDS samples after 3 days, 7 days and 28 days of curing, according to ASTM [37]. The broken specimens after UCS testing were crushed into powder and then weighed 20 g into a 100 mL beaker. Afterward, 50 mL of distilled water was gradually added and blended for 5 min. After 2 h, the pH and EC values were measured.

#### 2.3.3. Microstructural Analyses

Microstructural analyses of typical CPDS specimens after 7 days and 28 days of curing were also conducted by employing XRD and SEM tests. Before microstructural analyses, the specimens were immersed in ethanol for 7 days to inhibit hydration reactions before being freeze-dried in liquid nitrogen. Then, the specimens were vacuumed for 48 h to sublimate. SEM testing was performed on dried specimen pieces no larger than 7 mm, and the images of specimens were enlarged 20,000 and 50,000 times. XRD testing was performed on sample powder passing through 75 µm sieves, and the samples were scanned from 10° to 80° at a rate of 2°/min.

## 3. Results and Discussion

### 3.1. Unconfined Compressive Strength

#### 3.1.1. Effect of Cement Content

Figure 6 presents the effect of cement content on the UCS of CPDS at various CCR contents. It is evidently shown that the UCS of CPDS is effectively improved by the inclusion of cement under different CCR contents, namely, the UCS increases with the increase of cement content. For the 28-day cured samples, an almost linear relationship between UCS and cement content can be noted. Nonetheless, for the 3-day cured samples, the increments of UCS for the cement content increasing from 20% to 30% are higher than that for the cement content growing from 10% to 20% in a limited number of experiments. The result that increasing cement content can increase the UCS of CPDS was also confirmed by Xiao and Xu [38].

#### 3.1.2. Effect of CCR Content

To reduce cement consumption, CCR was incorporated into cement as an alkaline activator to improve the strength development of CPDS. Figure 7 shows the effect of CCR on the UCS of CPDS with different admixture conditions, and the error bars were used to illustrate the margin of error. It can be seen that the UCS increases with the initial increase in the CCR content; then, it declines when the CCR content exceeds the optimum value, except for 28-day cured specimens with 10% cement content. For example, for CPDS with 30% cement content, the optimal CCR content of CPDS cured for 3 days, 7 days and 28 days are 10%, 10% and 20%, respectively, and the maximum compressive strength values are 1068.2 kPa, 2277.5 kPa and 3348.6 kPa, which are 29.75%, 16.82% and 42.22% correspondingly higher than that of CPDS containing 30% cement content alone.

It can be noted that the CCR content effect is also related to curing time and cement amount. Figure 7b also shows that the slope of UCS curves versus CCR content reaches its maximum when the CCR content increases from 0% to 5% at 3 days and 7 days of curing time. Meanwhile, for the curing time of 28 days, when the CCR content increases from 10% to 15%, the slope of the curve reaches its maximum. That is, unreacted Ca(OH)_2_ may react with SiO_2_ and Al_2_O_3_ in the soil, and as the curing time goes on, the reaction consumes Ca(OH)_2_ persistently, leading to the formation of additional CSH gel. In addition, according to the UCS of CPDS with different cement contents cured for 28 days, the UCS of CPDS changes differently as the CCR content increases from 5% to 30%. Especially, the UCS of CPDS with 10% cement content increases continuously. Overall, for the CPDS with 10–30% cement content, the strength of CPDS cured for 28 days is significantly improved with the inclusion of 15–25% CCR content.

In order to further understand the beneficial effect of CCR content on the strength of CPDS, a strength growth ratio SIR was introduced in Equation (1) [38], and the results are shown in Table 4.
(1)$$\mathrm{SIR}=\frac{{\mathrm{UCS}}_{\left(C_{a}=0,5,10,15,20,25,30\%\right)}}{{\mathrm{UCS}}_{\left(C_{a}=0\%\right)}}$$
where C_a_ demotes the CCR content.

The SIR value is basically in the range of 0.72 to 1.42, and most of the SIR values are close to 1.00 or more than 1.00, which is consistent with the strength growth ratio of samples mixed steel slag powder (5%, 10%, 15%, 20%) with dredged sludge, as shown by Lang et al. [3]. Based on the obtained results, the maximum increments of UCS, viz. 244.87 kPa (R = 1.30), 327.97 kPa (R = 1.17), and 993.97 kPa (R = 1.42) are observed in the P30C10 specimen cured for 3 days and 7 days, and the P30C20 specimen cured for 28 days, respectively. Moreover, Table 4 shows that the continuous addition of CCR may have a detrimental effect on the strength development of CPDS, showing that when CCR content is excessive, the strength does not increase but rather decreases, resulting in SIR less than 1.0 correspondingly, which is consistent with Figure 7. Additionally, the curing time is also crucial to the strength growth of CPDS, and the importance of curing time on CPDS strength development will be discussed in detail later.

As evident, using CCR as a solid alkaline activator for cemented dredged sludge is effective for improving the unconfined compressive strength. CaO and Ca(OH)_2_ are abundant in both dry and dissolved CCR, providing the environment of Pozzolanic reaction and formation of hydration products [39]. In this context, the additional production of CSH and CAH gels is associated to the strength development of CPDS. However, the continuous increase in CCR content decreases the strength, as shown in Figure 7. The potential causes will be investigated further using XRD and SEM analyses.

#### 3.1.3. Effect of Curing Age

It is evident that the maintenance time significantly influences the strength growth of CPDS. In the present experiments, the change of strength increment was used from 3 days to 7 days and 7 days to 28 days, respectively. The evolution of the UCS increment of CPDS from 7 days to 28 days under different admixture conditions (CCR and cement content) reflects the long-term gain of CPDS strength. A strength difference (ΔUCS) and a strength growth rate (UCS_gr_) [3] were introduced in Equations (2)–(5) to investigate the optimum CCR content for the long-term strength gain of CPDS, and the results at 30% cement content are given in Figure 8. The rate of increase in UCS for 7–28 days curing specimens is evidently slower than that for 3–7 days curing specimens, which demonstrates that the curing occurs primarily in the first 7 days, which is consistent with the previous research results.
(2)$$∆{\mathrm{UCS}}_{3-7}={\mathrm{UCS}}_{\left(7\right)}-{\mathrm{UCS}}_{\left(3\right)}$$
(3)$${\mathrm{UCS}}_{\mathrm{gr}3-7}=\frac{{\mathrm{UCS}}_{\left(7\right)}-{\mathrm{UCS}}_{\left(3\right)}}{{\mathrm{UCS}}_{\left(3\right)}}=\frac{∆\mathrm{UCS}}{{\mathrm{UCS}}_{\left(3\right)}}$$
(4)$$∆{\mathrm{UCS}}_{7-28}={\mathrm{UCS}}_{\left(28\right)}-{\mathrm{UCS}}_{\left(7\right)}$$
(5)$${\mathrm{UCS}}_{\mathrm{gr}7-28}=\frac{{\mathrm{UCS}}_{\left(28\right)}-{\mathrm{UCS}}_{\left(7\right)}}{{\mathrm{UCS}}_{\left(7\right)}}=\frac{∆\mathrm{UCS}}{{\mathrm{UCS}}_{\left(7\right)}}$$
where ΔUCS represents the strength difference, kPa; UCS_(3)_, UCS_(7)_ and UCS_(28)_ are the UCS of CPDS cured for 3 days, 7 days and 28 days, respectively, kPa; and UCS_gr_ represents the strength growth rate, %.

Moreover, the ΔUCS and UCS_gr_ were also used to assess the impact of maintenance time on the CPDS strength gain of all samples. Figure 9a illustrates the ΔUCS of CPDS vs. CCR content at various cement contents, and Figure 9b illustrates the UCS_gr_ of CPDS cured for 28 days. As shown in Figure 9a, the ΔUCS_(7–28)_ of CPDS with 30% cement content is obviously larger than that of CPDS with 10% and 20% cement content, meaning that the mixture of CCR activator and higher cement content is more beneficial to improve the strength development of CPDS. Figure 9a also demonstrates that except for 10% cement, the ΔUCS_(7–28)_ of CPDS with 20% and 30% cement increases initially and then decreases as the CCR content increases, reaching the maximum with 20% CCR, which is consistent with the results mentioned above.

Moreover, except for P10C10 having relatively low strength, it could be observed from Figure 9b that the UCS_gr(7–28)_ of CPDS with 20% and 30% cement content is more obvious than the UCS_gr(7–28)_ of CPDS with 10% cement content. The UCS_gr(7–28)_ values of P20C20 and P30C20 reach the maximum values of 82.60% and 68.53%, respectively. This means that for both 20% cement and 30% cement, the incorporation of 20% CCR is the most favorable for long-term strength development. Thus, it is recommended to use 20% cement admixed with 20% CCR to solidify dredged sludge.

### 3.2. pH and Electric Conductivity

The alkaline environment has a significant effect on the strength development of the cemented soil [3]. Therefore, in this study, the correlation between pH, EC values and UCS of CPDS cured for 7 days with and without the addition of activator were investigated. The relationships between pH, EC values and UCS versus cement content are presented in Figure 10. It is clearly observed that a positive association exists between the pH, EC and UCS values as the cement content increases. This observation confirmed that an alkaline environment is conducive to the cemented dredged sludge strength gain, which contributed to the presence of Ca(OH)_2_ generated during cement hydration.

Figure 11 gives the influence of CCR content on the relationship between pH, EC and UCS of CPDS. As shown, adding CCR increases pH, EC and strength linearly firstly, indicating the effectiveness of increasing CCR content on the strength development of CPDS. It is well known that SiO_2_ and Al_2_O_3_ in soil particles are essentially weak acidic, and they will gradually dissolve in a strong alkaline environment and react with Ca(OH)_2_ to generate hydration products such as CAH and CSH gels. Simultaneously, the incorporation of CCR increases the Ca(OH)_2_ content in the solution, not only stimulating more SiO_2_ and Al_2_O_3_ in the soil particles to participate in the hydration reaction, but also more hydration reaction and volcanic ash reaction can be carried out and mutually enhanced, which then significantly improves the strength of CPDS. However, as the CCR content exceeds a certain value, an inverse relationship is found between pH, EC values, and UCS. The observed relationship might be explained as follows: the continuous addition of CCR will result in an increase in the concentration, and then, excessive OH^−^ concentration will hinder the dissolution of Ca(OH)_2_. Furthermore, if the Ca(OH)_2_ concentration is excessive, the thickening of alkali solution will reduce ion mobility such as Si^4+^ and Al^3+^ from the soil particles, preventing further leaching out of ions; similar phenomena have been identified by other researchers [40,41]. Huo et al. [41] investigated the variation of Si^4+^ and Al^3+^ ions precipitation concentration in NaOH solution of different concentrations, and they found that the precipitation concentration of Si^4+^ and Al^3+^ ions increased with the increase in NaOH solution concentration. However, when the concentration of NaOH solution further increased, the concentration of Si^4+^ and Al^3+^ ions in solution decreased. Thus, these results suggest that a too little or too high concentration of alkali solution inhibits the hydrolysis and precipitation of Si^4+^ and Al^3+^ ions from soil particles, as well as the dissolution of Ca(OH)_2_, further preventing the volcanic ash reaction and the development in strength. Together, these findings demonstrate that CPDS with a high alkalinity does not necessarily achieve a high UCS, and there is an optimal alkaline environment for the strength gain of CPDS.

The average pH value of 7.2 was recorded for dredged sludge, whereas the pH values corresponding to the maximum UCS of CPDS cemented by 10%, 20%, and 30% content are 9.9, 10.1, and 10.5 correspondingly. In this study, the pH value of 9.9–10.5 is appropriate for the strength development of CPDS, which is consistent with the previous studies of cement-SSP-stabilized dredged sludge (pH value of 9.9–10.0) [3]. The above observations suggest that the proper inclusion of CCR plays a positive role in providing the alkaline environment, which could be a crucial index to assess the strength gain of CPDS.

### 3.3. SEM Analysis

#### 3.3.1. Effect of CCR Content

To investigate the microstructure variation and compositional evolution of CPDS and reveal the contribution of CCR to strength growth, P20C0, P20C10, P20C20 and P20C30 specimens cured for 7 days and 28 days were selected as typical examples for SEM testing, and the results are shown in Figure 12. The SEM images of the P20C0 specimen show that soil particles were coated with few gels, and the lap joints between gels were weak. In addition, a large amount of macropores with honeycomb and irregular shape still existed between soil particles, indicating that very limited hydration occurred.

Figure 12b–d presents the morphology of CPDS with 20% cement content admixed with 10%, 20%, and 30% CCR, respectively. With the addition of CCR, the fibrous and flocculent colloids in the structural morphology increased significantly, which led aggregates to bond with each other more closely, and more cubic-shaped CAH particles were adsorbed on the surface of the aggregates. As a result, a more homogeneous and compact formation of gels gave a positive effect on the UCS, which was consistent with the results obtained from the above mechanical analyses.

In addition, Figure 12d shows P20C30 specimens having fewer internal pores compared to P20C20 specimens, and it may result in an improvement in mechanical properties. However, the reverse results were obtained by the UCS tests (as shown in Figure 7), which might be due to the difference in CSH morphology. It is well known that the pozzolanic reaction produces a variety of CSH gels depending on the concentration of Ca^2+^ [42]. The CSH crystal morphology of P20C30 specimens maintained for 28 days was different from that of P20C20 specimens, as the former had fewer fibrous CSH and changed to large amounts of small-sized rounded CSH (as shown in Figure 13). A higher CCR content leads to a higher Ca/Si ratio of CSH, and the binding capacity of CSH decreases with increasing Ca/Si ratio [43,44], which may result in a lower strength of P20C30 specimens than P20C20 specimens. Furthermore, it has been observed that the high lime content in slag produces α-dicalcium silicate hydrate (α-C2SH), which has a high Ca/Si ratio and contributes to reduced strength [45].

Some interesting phenomena were also found in these SEM images; namely, a small amount of sheet-like crystals of Portlandite crystals were found in 7-day P20C20 specimens, 7-day P20C30 specimens, and 28-day P20C30 specimens (as shown in Figure 12). A possible explanation for this might be that the excessive admixture of CCR resulted in the transformation of Ca(OH)_2_ from a saturated solution to a supersaturated solution and then turned into Portlandite crystals. These single sheets of Portlandite crystals had apparent hexagons and were stacked on the surface of gels, forming a weakness in the matrix and affecting strength growth, matching the phenomenon found by Sun et al. [46]. As a result, the above observations can further demonstrate the effect of CCR to enhance the strength of CPDS and also indicate that the excessive addition of CCR will have a negative impact on the strength increase.

#### 3.3.2. Effect of Curing Time

Curing time has a significant effect on the strength development of CPDS due to the chemical cementation action. Taking P20C0, P20C10, P20C20, and P20C30 specimens as typical examples (shown in Figure 12), the impacts of chemical cementation on the strength development of the CPDS as curing time increases are investigated. The SEM images show that the cementitious products obviously increased as the maintenance time increased from 7 to 28 days, and the cementation bond controls the strength development [47], confirming that prolonging the curing time led to a higher strength. Meanwhile, the space of the voids between the clusters reduced in specimens cured for 28 days.

### 3.4. XRD Analysis

To further study the impacts of CCR content and maintenance time on the hydration products, XRD tests were carried out on P20C0, P20C10, P20C20 and P20C30 specimens, and the XRD patterns are displayed in Figure 14. It was shown that all samples contained amorphous CSH and cubic-shaped CAH, indicating that CSH and CAH are the primary hydration products. In addition, the characteristic peak of CaCO_3_ was also noticed in all samples. Aside from the small amount in the CCR, the remaining CaCO_3_ was produced by the reaction between Ca^2+^ on and CO_2_ in the atmosphere [48]. Moreover, Gismondine gels were formed as a result of the reaction of Ca(OH)_2_ with reactive pozzolanic materials [49]:(6)$$\mathrm{Ca}{\left(\mathrm{OH}\right)}_{2}+{\mathrm{Al}}_{2}O_{3}+{\mathrm{SiO}}_{2}+H_{2}O\to {\mathrm{CaAl}}_{2}{\mathrm{Si}}_{2}O_{8}\cdot 4H_{2}O$$

The production of Gismondine is conducive to enhancing the strength development of treated soil, which has been proved before [50]. Interestingly, the characteristic peak of ettringite (AFt) was detected in P20C20 and P20C30 specimens, certifying that the needle-like gel identified in SEM tests was AFt. CCR contains a certain amount of $\;{\mathrm{SO}}_{4}^{2-}$ ions, which are very low in number but occupy a crucial role and are an important source of AFt production, so AFt was detected when the incorporation of CCR increased, as described by the following reaction [51]:(7)$$\mathrm{Ca}{\left(\mathrm{OH}\right)}_{2}+{\mathrm{Al}}_{2}O_{3}+{\mathrm{CaSO}}_{4}+H_{2}O\to 3\mathrm{CaO}\cdot {\mathrm{Al}}_{2}O_{3}\cdot 3{\mathrm{CaSO}}_{4}\cdot \left(30-32\right)H_{2}O$$

In contrast, the Portlandite (Ca(OH)_2_) phase still survived in the XRD patterns of P20C30 specimens, which was also confirmed by the SEM analyses as discussed earlier, indicating that the CCR mixed into the mixtures is not completely consumed by the hydration reaction. This finding is contrary to previous studies which have found the absence of Ca(OH)_2_ in CCR-GGBS and lime-GGBS stabilized soils [34,52]. A possible explanation for this might be that the limited solubility of Ca(OH)_2_ solution makes only a small fraction of Ca(OH)_2_ dissolved in alkaline solutions [53], and when it exceeds the limited concentration under the high alkaline solutions, the Ca(OH)_2_ will occur [42].

The Suolunite phase was observed in P20C20 and P20C30 specimens after curing of 28 days, which could not be found in other specimens. Suolunite, as a high hardness double tetrahedral crystal, benefits the strength development compared with the crystal structure of synthesized CSH gels in cement hydration products [54]. It is reasonable to deduce that the existence of Suolunite crystal enhances the strength improvement in P20C20 and P20C30 specimens to some extent.

### 3.5. Carbon Footprint Evaluation

The carbon footprint, defined as CO_2-e_ emitted (kg CO_2-e/ton_) [55], of CPDS was calculated and compared with that of cemented dredged sludge, which reached the same level of strength in Figure 15. The CO_2_ emission was calculated, taking into account the exploiting, processing and producing of raw materials. The values of CCR and cement are 0.007 kg and 0.86 kg CO_2-e/ton_, respectively [55]. As shown in Figure 15, the emission CO_2-e_ values of CPDS specimens (P20C15, P20C20, P20C25, P20C30) were 95.2, 86.37, 88.62 and 100.35 kg CO_2-e/ton_ for the same UCS of 800 kPa, respectively, while the emission CO_2-e_ value was 106.78 kg CO_2-e/ton_ for cemented dredged sludge. Table 5 shows the emission CO_2-e_ value of CCR and cement, respectively, and the reduction of CPDS comparing to cemented dredged sludge. It is apparent from this table that the emission CO_2-e_ values of P20C15, P20C20, P20C25 and P20C30 were 10.84%, 19.11%, 17.01% and 6.02% lower than that of cemented dredged sludge at the same UCS of 800 kPa, and the maximum was up to 19.11%. At this point, the carbon footprint is significantly reduced with the addition of CCR, and under the optimal CCR admixture, CPDS can not only improve the mechanical strength but also reduce the cement consumption and CO_2_ emissions, thus achieving sustainable soil improvement from both engineering and environmental perspectives.

### 3.6. Solidification Mechanism

The above analyses show that the strength development of solidified dredged sludge depends mainly on the amount and morphology of the cementitious products. These products are mainly produced by a series of physical and chemical reactions between cement, CCR and dredged sludge, mainly including hydration reactions, ion exchange, volcanic ash reactions, and carbonation reactions [56]. To clarify, the schematic of the mechanism for CPDS is shown in Figure 16.

The strength of cement–soil mainly comes from the cementation of two parts of hydrates; one part is the cementation of hydration products of cement itself, and the other part is the cementation of hydrates produced by the volcanic ash reaction between Ca(OH)_2_ provided by cement hydration and active materials in the soil. The former constitutes the main part of the strength of cement–soil. When CCR is mixed into the cement–soil as an alkaline activator, firstly, CaO reacts with water to produce a large amount of Ca(OH)_2_ (reaction Formula (8)), which ensures the sufficient Ca(OH)_2_ in the pore water after soil absorbs Ca^2+^, OH^−^ and CaO in large quantities in the early stage. Secondly, with the increase in Ca/Si, OH^−^/Si, the proportion of CSH in cement hydration products will increase. As the hydration reaction proceeds, more CAH, CSH, Gismondine, and other hydration products (reaction Formulas (9)–(11)) will be formed from the reaction between free SiO_2_ and Al_2_O_3_ on the surface of soil particles with Ca(OH)_2_, which makes the structure denser and improves the overall strength. In addition, free Ca(OH)_2_ in pore water can absorb CO_2_ from air and water, and carbonate reaction occurs to form water-insoluble CaCO_3_ precipitates (reaction Formula (12)), which cannot provide strength directly but can fill the pores between soil particles and further improve the strength of the soil. Moreover, for the CPDS with high CCR content, AFt crystals (reaction Formula (13)) with certain swelling effects (due to the presence of ${\mathrm{SO}}_{4}^{2-}$ in CCR) and Suolunite crystals with high hardness will be produced. These crystals and hydration products are interwoven to continuously fill the pores between soil particles, resulting in a denser structure.

The above analyses show that CCR as an alkaline exciter can effectively improve the strength of the cemented dredged sludge. However, the strength will tend to decrease with excessive CCR, which may be due to the following reasons:

(1) The binding capacity of CSH decreases with the increase in Ca/Si. When Ca/Si in pore water is too high, the morphology of CSH also appears to be different, which is confirmed in SEM.

(2) There is a threshold value of Ca(OH)_2_ concentration, and when the Ca(OH)_2_ concentration reaches a certain value that exceeds its threshold, the saturated solution will turn to a supersaturated solution, which will lead to the precipitation of Ca(OH)_2_ (confirmed in SEM, XRD). The precipitation of Ca(OH)_2_ increases the concentration of the solution, and it also hinders the further dissolution and transport of Ca^2+^ and OH^−^. The Ca(OH)_2_ precipitation on the surface of soil particles further hinders the dissolution of silica–alumina active components in soil particles, which inhibits the volcanic ash reaction and leads to a decrease in the amount of hydration products.

(3) The Ca(OH)_2_ precipitates as a monolayer structure, which promotes the formation of a weak strength surface and leads to a decrease in strength.

The relevant chemical reactions among the cement and CCR are summarized here:(8)$$\mathrm{CaO}+H_{2}O\to \mathrm{Ca}{\left(\mathrm{OH}\right)}_{2}$$
(9)$$\mathrm{Ca}{\left(\mathrm{OH}\right)}_{2}+{\mathrm{SiO}}_{2}+H_{2}O\to \mathrm{xCaO}\cdot {\mathrm{ySiO}}_{2}\cdot {\mathrm{zH}}_{2}O$$
(10)$$\mathrm{Ca}{\left(\mathrm{OH}\right)}_{2}+{\mathrm{Al}}_{2}O_{3}+H_{2}O\to \mathrm{xCaO}\cdot {\mathrm{yAl}}_{2}O_{3}\cdot {\mathrm{zH}}_{2}O$$
(11)$$\mathrm{Ca}{\left(\mathrm{OH}\right)}_{2}+{\mathrm{Al}}_{2}O_{3}+{\mathrm{SiO}}_{2}+H_{2}O\to {\mathrm{CaAl}}_{2}{\mathrm{Si}}_{2}O_{8}\cdot 4H_{2}O$$
(12)$$\mathrm{Ca}{\left(\mathrm{OH}\right)}_{2}+{\mathrm{CO}}_{2}\to {\mathrm{CaCO}}_{3}+H_{2}O$$
(13)$${\mathrm{Al}}_{2}O_{3}+{\mathrm{Ca}}^{2+}+{\mathrm{OH}}^{-}+{\mathrm{SO}}_{4}^{2-}\to 3\mathrm{CaO}\cdot {\mathrm{Al}}_{2}O_{3}\cdot 3{\mathrm{CaSO}}_{4}\cdot 32H_{2}O$$

## 4. Conclusions

The effect of calcium carbide residue (CCR) on the strength development and micro-mechanisms of CCR–Portland cement-stabilized dredged sludge (CPDS) were evaluated. The roles of CCR content, cement content and curing time on strength gain were investigated. Several important findings can be obtained in this work:CCR improved the UCS of CPDS significantly, and the maximum strength of CPDS was 1.42 times that of cemented dredged sludge cured for 28 days. However, the UCS then decreased as CCR content increased continuously, suggesting that excessive CCR may have a detrimental effect on the strength development of CPDS.On the basis of the strength difference (ΔUCS) and strength growth rate (UCS_gr_) of CPDS, it was recommended that the usage of 20% cement with 20% CCR to solidify dredged sludge is the most effective to develop the long-term strength.An alkaline environment promotes the development of CPDS strength, and the strength increased initially and then decreased with the growth of pH. Overall, the pH value corresponding to UCS of 9.9–10.5 may be suitable for evaluating the strength gain of CPDS.The SEM and XRD results indicated that the addition of CCR promoted the formation of hydration products and the formation of denser structure in CPDS. However, the excessive amount of CCR would lead to the precipitation of Ca(OH)_2_ and the change of CSH morphology, resulting in a risk of strength loss.The emission CO_2-e_ of CPDS (20% cement content with the addition of 20% CCR) was 19.11% lower than that of cemented dredged sludge at UCS of 800 kPa, showing that the carbon footprint was significantly reduced with the addition of CCR. Under the optimal CCR admixture, CPDS can not only improve the mechanical properties but also reduce the cement consumption and CO_2_ emissions, thus achieving sustainable soil improvement from both engineering and environmental perspectives.

It can be concluded that the CCR is effective as a substitute alkaline activator of Portland cement-stabilized dredged sludge. In addition, the CCR is by-products, and it can achieve the goal of waste utilization and reduction of cement consumption at the same time. Therefore, this study provides a substitute alkaline activator for stabilized dredged sludge.

## Figures and Tables

**Figure 1 materials-15-04453-f001:**
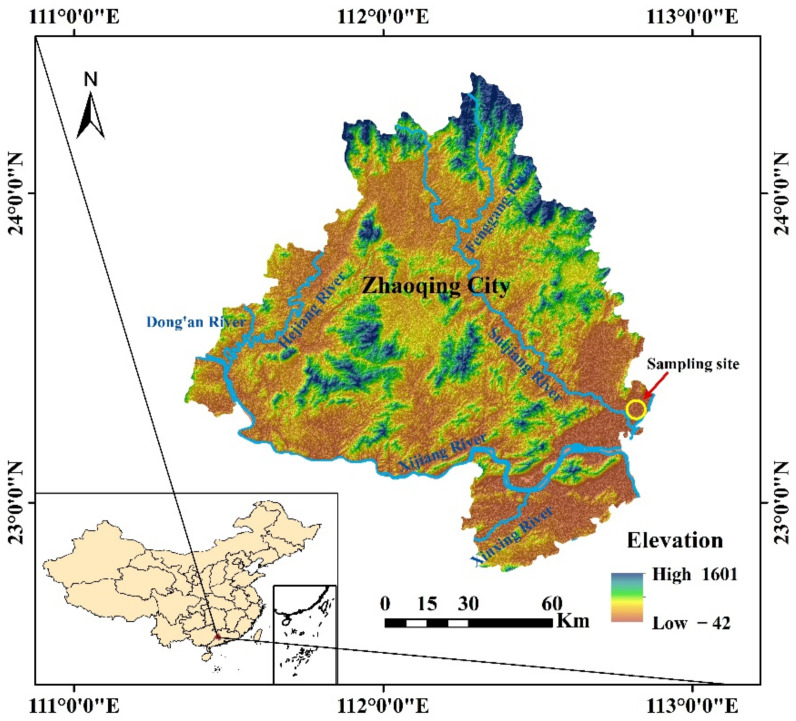
Location of dredged sediment sludge.

**Figure 2 materials-15-04453-f002:**
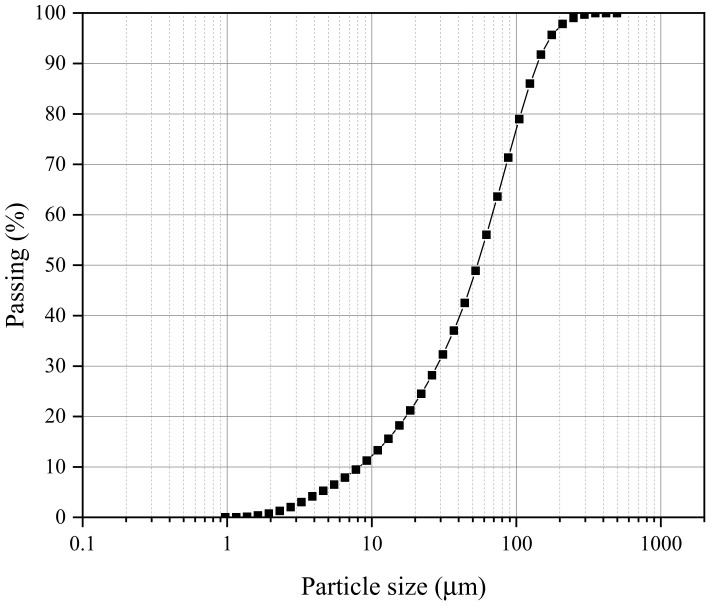
Particle size distribution of dredged sludge.

**Figure 3 materials-15-04453-f003:**
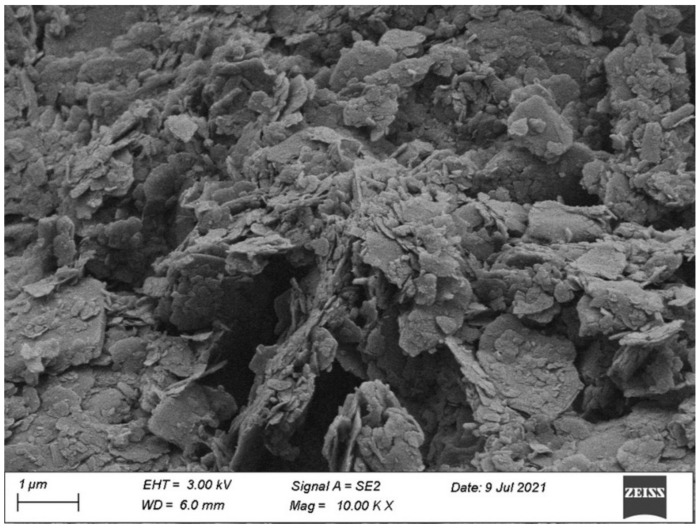
The microstructure of dredged sludge.

**Figure 4 materials-15-04453-f004:**
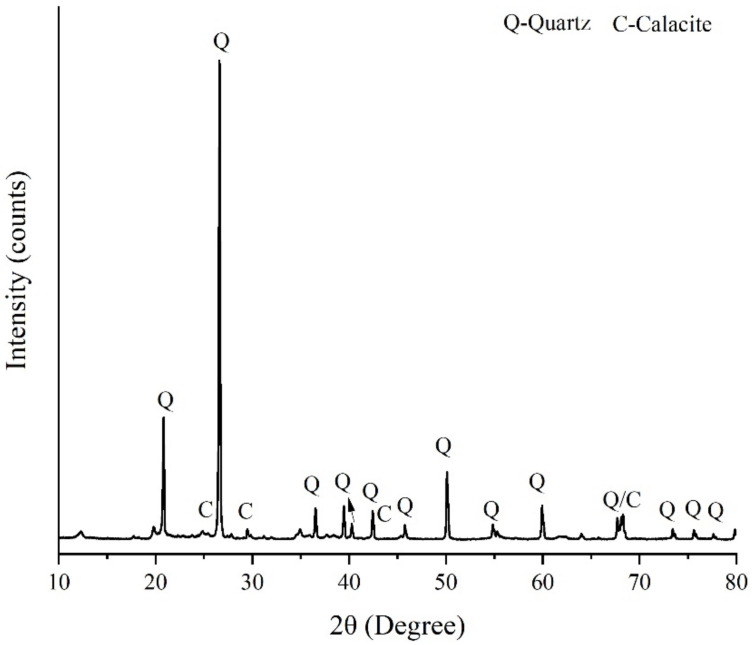
X-ray diffraction patterns of dredged sludge.

**Figure 5 materials-15-04453-f005:**
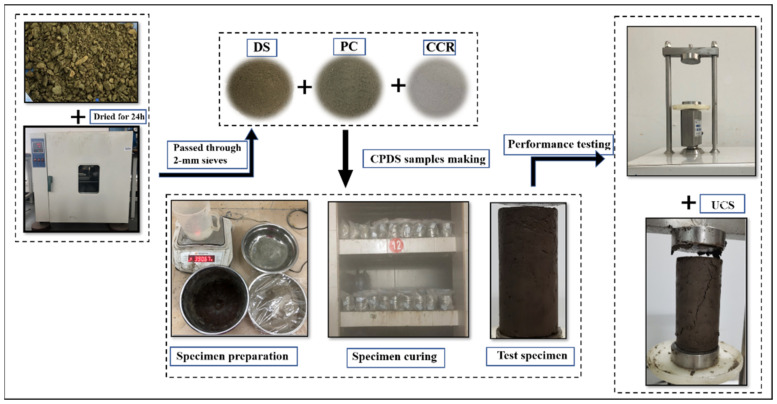
The preparation process of CPDS.

**Figure 6 materials-15-04453-f006:**
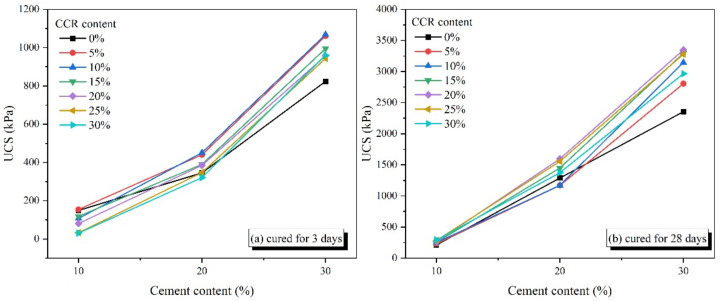
Effect of cement content on the UCS of CPDS cured for (**a**) 3 days and (**b**) 28 days under different CCR contents.

**Figure 7 materials-15-04453-f007:**
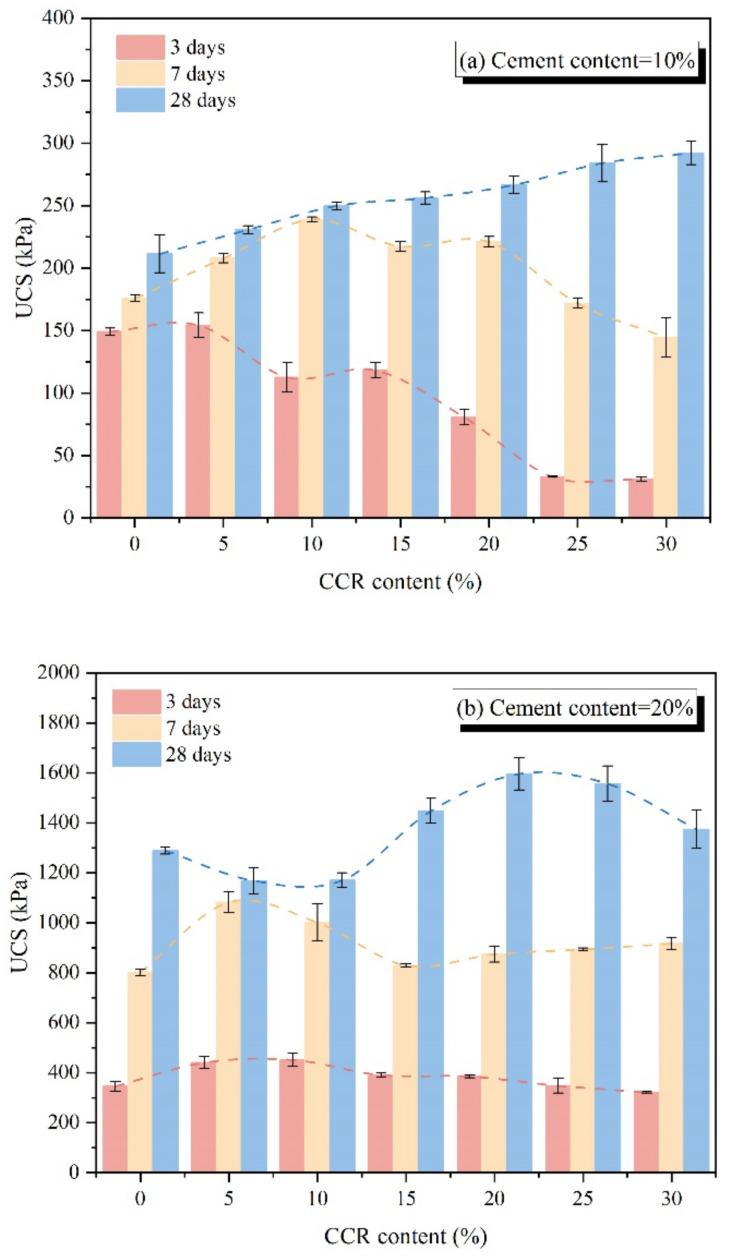
Effect of CCR content on the UCS of CPDS at (**a**) cement content = 10%, (**b**) cement content = 20% and (**c**) cement content = 30%.

**Figure 8 materials-15-04453-f008:**
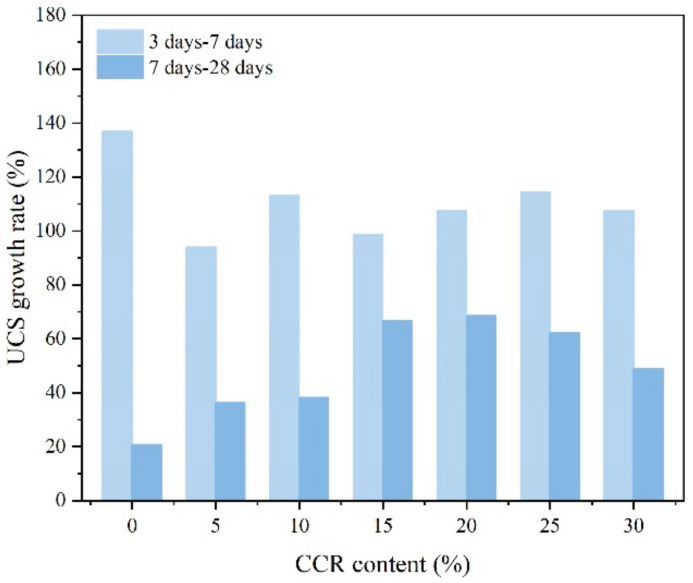
The UCS_gr(3–7)_ and UCS_gr(7–28)_ of CPDS versus the CCR content at 30% cement content.

**Figure 9 materials-15-04453-f009:**
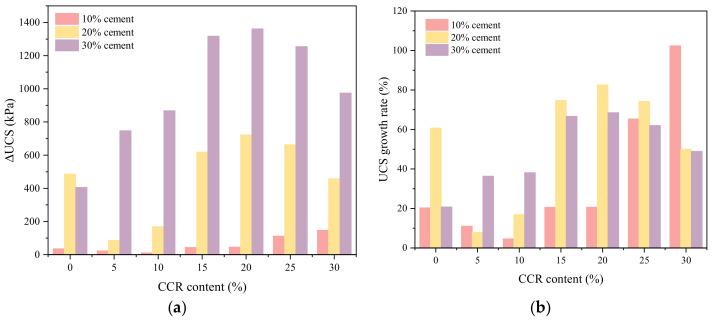
The (**a**) ΔUCS_(7–28)_ and (**b**) UCS_gr(7–28)_ of CPDS versus the CCR content at different cement contents.

**Figure 10 materials-15-04453-f010:**
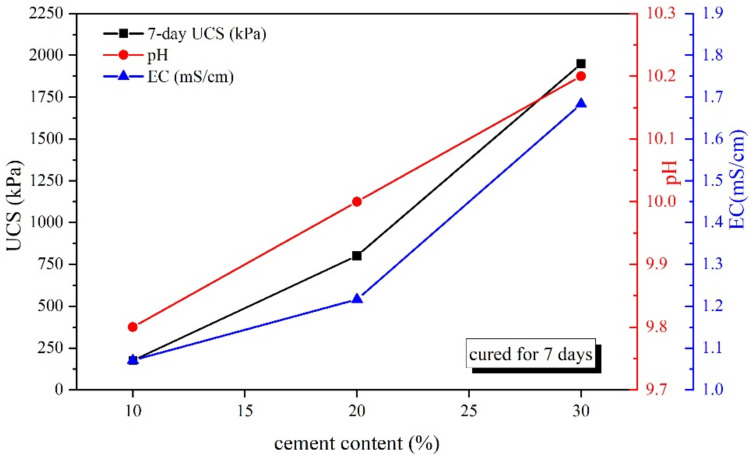
Relationships between cement content and UCS, pH and EC of CPDS cured for 7 days.

**Figure 11 materials-15-04453-f011:**
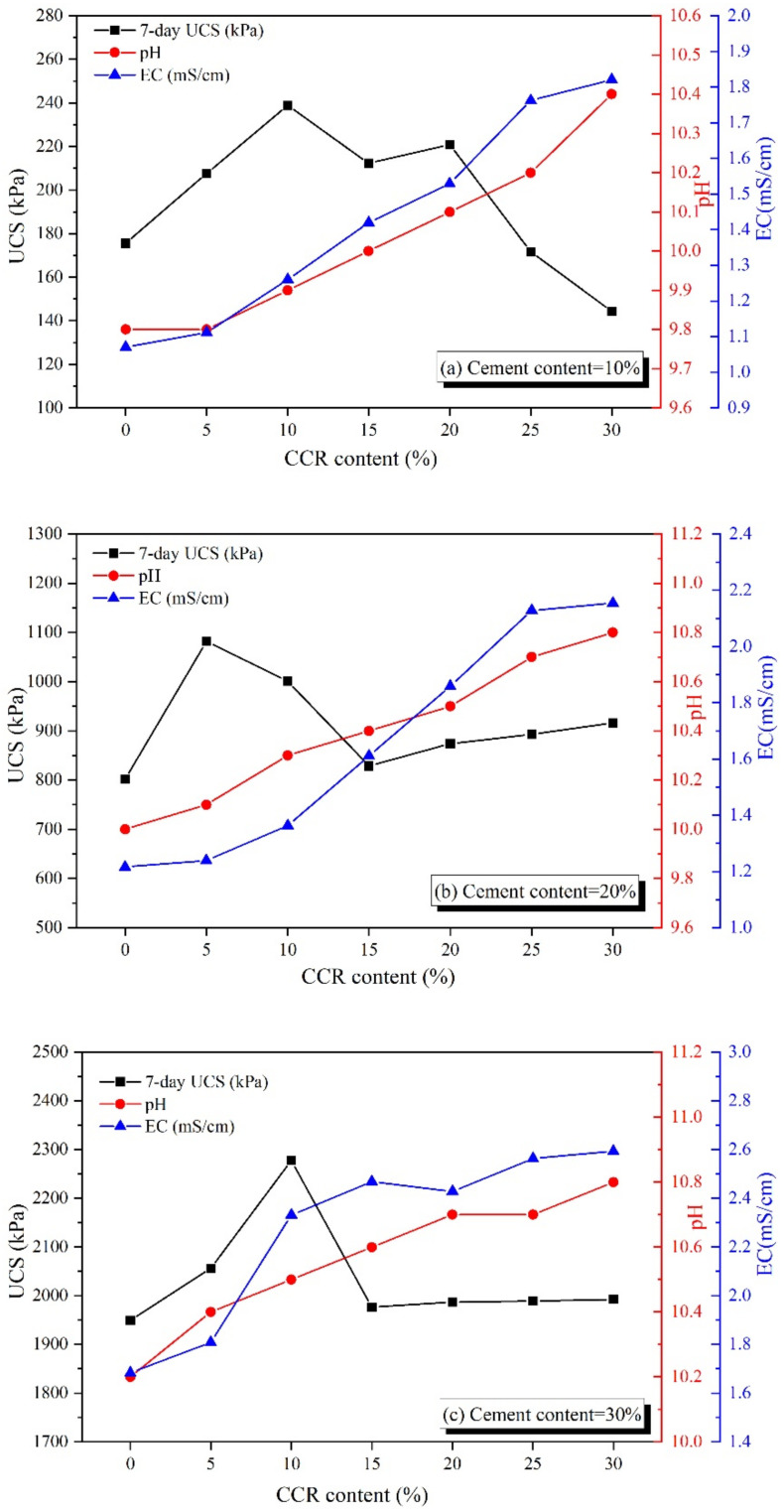
Relationships between CCR content and UCS, pH and EC of CPDS cured for 7 days with (**a**) cement content = 10%, (**b**) cement content = 20% and (**c**) cement content = 30%.

**Figure 12 materials-15-04453-f012:**
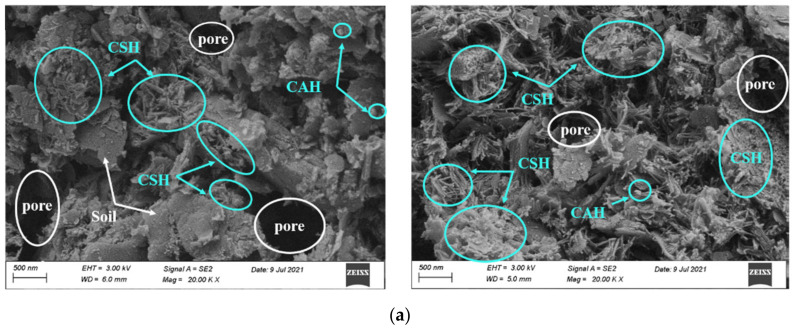
SEM images of CPDS specimens cured for 7 days (the **left**) and 28 days (the **right**). (**a**) Specimens P20C0; (**b**) Specimens P20C10; (**c**) Specimens P20C20; (**d**) Specimens P20C30.

**Figure 13 materials-15-04453-f013:**
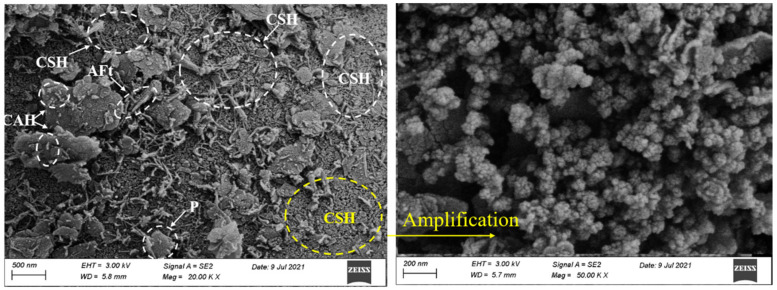
SEM images of CSH gels.

**Figure 14 materials-15-04453-f014:**
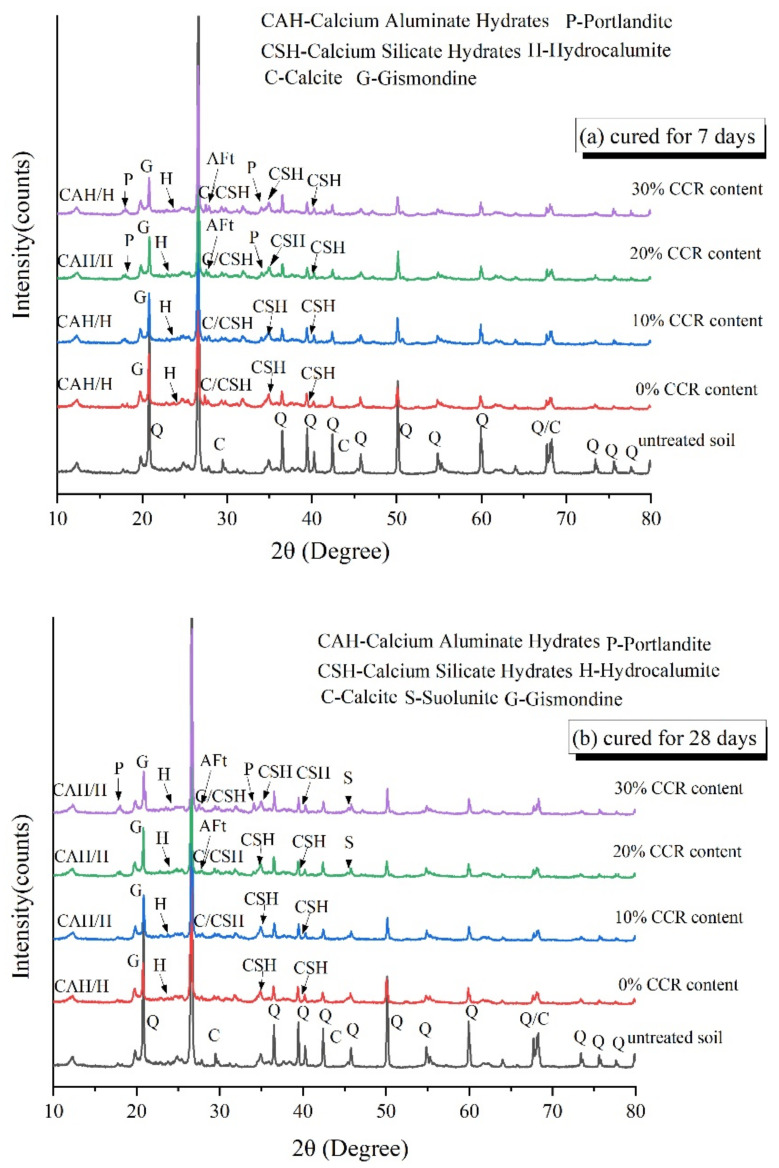
X-ray diffraction patterns of untreated and CCR stabilized soil cured for: (**a**) 7 days and (**b**) 28 days with 20% cement content.

**Figure 15 materials-15-04453-f015:**
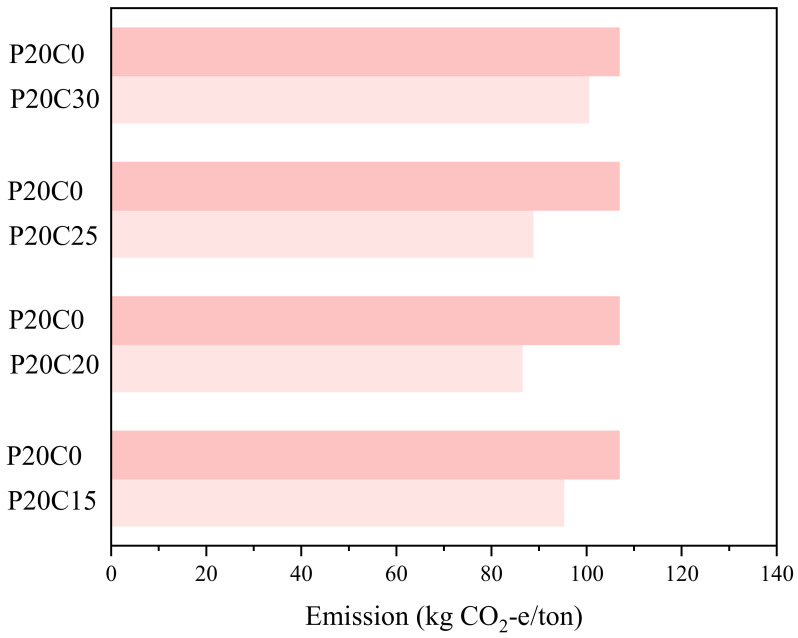
Total CO_2-e_ of CPDS (P20C15, P20C20, P20C25 and P20C30) and cemented dredged sludge.

**Figure 16 materials-15-04453-f016:**
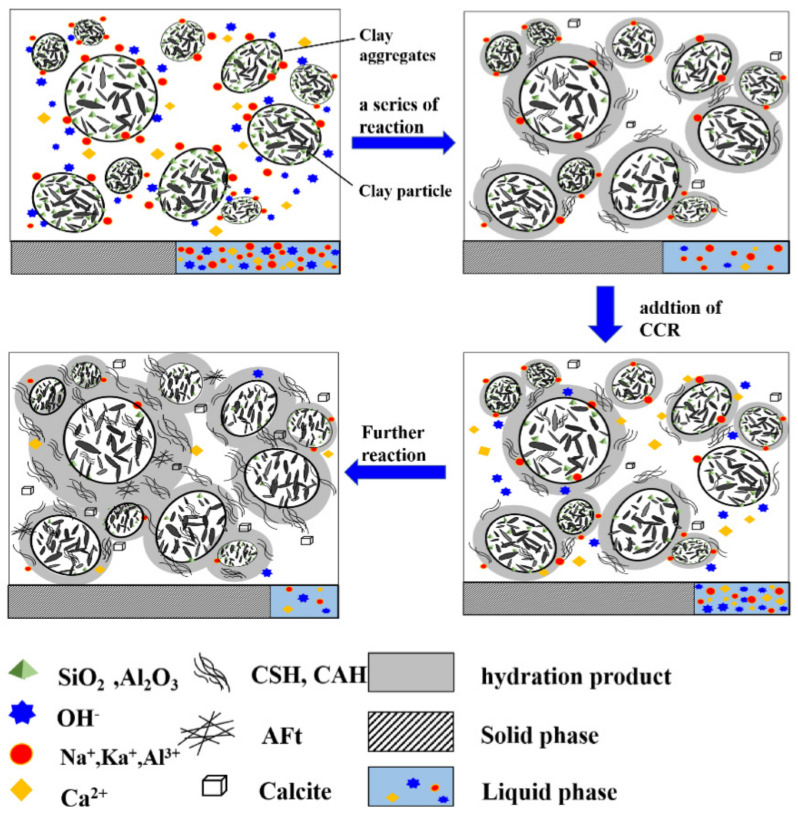
The solidification mechanism of CPDS.

**Table 1 materials-15-04453-t001:** The basic physical properties of dredged sludge.

Properties	Values
Initial water content, %	60.0
Liquid limit, %	50.0
Plastic limit, %	27.6
Plasticity index, %	22.4
Dry density, g/cm^3^	1.01
pH	7.20
Electrical conductivity (EC), mS/cm	0.0391

**Table 2 materials-15-04453-t002:** Chemical compositions of dredged sludge, CCR and Portland cement.

Oxide (%)	CaO	SiO_2_	Al_2_O_3_	Fe_2_O_3_	MgO	SO_3_	Na_2_O	K_2_O	Others
Dredged sludge	0.602	57.276	27.417	8.555	1.609	0.329	0.360	2.387	1.464
CCR	94.297	2.654	0.836	0.262	0.159	0.446	1.045	-	0.301
Portland cement	59.450	20.940	6.661	2.718	4.225	4.208	0.189	0.924	0.686

**Table 3 materials-15-04453-t003:** Mixture design of the samples.

Sample	PC Content	CCR Content	Sample	PC Content	CCR Content	Sample	PC Content	CCR Content
	(%)	(%)		(%)	(%)		(%)	(%)
P10C0	10	0	P20C0	20	0	P30C0	30	0
P10C5	10	5	P20C5	20	5	P30C5	30	5
P10C10	10	10	P20C10	20	10	P30C10	30	10
P10C15	10	15	P20C15	20	15	P30C15	30	15
P10C20	10	20	P20C20	20	20	P30C20	30	20
P10C25	10	25	P20C25	20	25	P30C25	30	25
P10C30	10	30	P20C30	20	30	P30C30	30	30

**Table 4 materials-15-04453-t004:** The strength increase ratio of CPDS prepared by different cement and CCR content.

Sample	SIR			Sample	SIR			Sample	SIR		
	3 Days	7 Days	28 Days		3 Days	7 Days	28 Days		3 Days	7 Days	28 Days
P10C0	1.00	1.00	1.00	P20C0	1.00	1.00	1.00	P30C0	1.00	1.00	1.00
P10C5	1.03	1.18	1.09	P20C5	1.27	1.35	0.91	P30C5	1.29	1.06	1.19
P10C10	0.72	1.36	1.18	P20C10	1.30	1.25	0.91	P30C10	1.30	1.17	1.34
P10C15	0.79	1.21	1.21	P20C15	1.13	1.03	1.12	P30C15	1.21	1.01	1.40
P10C20	0.54	1.26	1.26	P20C20	1.11	1.09	1.24	P30C20	1.16	1.02	1.42
P10C25	0.22	0.98	1.34	P20C25	1.00	1.11	1.21	P30C25	1.15	1.04	1.39
P10C30	0.21	0.82	1.38	P20C30	0.93	1.14	1.07	P30C30	1.17	1.02	1.26

**Table 5 materials-15-04453-t005:** The emission and reduction of CPDS for per ton of dry soil at 800 kPa strength.

Emission	CPDS				Cemented Dredged Sludge
	P20C15	P20C20	P20C25	P20C30	P20C0
Cement (kg CO_2-e/ton_)	95.09	86.23	88.44	100.11	106.78
CCR (kg CO_2-e/ton_)	0.12	0.14	0.18	0.24	/
Total (kg CO_2-e/ton_)	95.20	86.37	88.62	100.35	106.78
Reduction (%)	10.84	19.11	17.01	6.02	/

## Data Availability

All the data in the tests of this study have been listed in this paper.

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
