# Peer review of "Effect of Calcium Carbide Residue on Strength Development along with Mechanisms of Cement-Stabilized Dredged Sludge"

_materials, 2022, doi:10.3390/ma15134453_

Round 1

Reviewer 1 Report

I find that the paper entitled: "Effect of calcium carbide residue on strength development along with mechanisms of cement-stabilized dredged sludge" is an interesting.

 The paper deals with research of feasibility of using calcium carbide residue (CCR), a by-product from acetylene gas production, as a solid alkaline activator on the strength development in CCR-Portland cement-stabilized dredged sludge (CPDS).

Based on the detailed results presented by the authors, it can be concluded that the CCR is effective as a substitute alkaline activator of Portland cement-stabilized dredged sludge. Additionally, the CCR is by-products and it can achieve the goal of waste utilization and reduction of cement consumption at the same time. Thus, this study provides a substitute alkaline activator for stabilized dredged sludge.

The list of references is up-to-date and more than enough.

I recommend the authors to review the work technically and eliminate text errors.

 I recommend the authors to insert image tags such as (a), (b), еtc.  into the image, not below the image, and to present in Figure captions a clear description of what is in picture (a), (b), etc...

Accordingly, I recommend accept after minor revision (corrections to minor methodological errors and text editing).

Reviewer 2 Report

After reviewing, this research can be published after major revision as follows:

1. Rewrite "Fi- 23 nally, the solidification mechanism of CPDS was discussed and revealed. Accordingly, it was con- 24 firmed that CCR can be a sustainable alternative solid alkaline activator to traditional alkaline acti- 25 vators for the aim of improving cemented dredged sludge in terms of environmentally friendly. 26"

2. This is a good conclusion identifying the novelty "Therefore, the aim of this study is to inves- 95 tigate the effect of CCR as a sustainable activator to improve mechanical properties of 96 cement-stabilized dredged sludge. A variety of tests including unconfined compressive 97 strength (UCS), pH and electric conductivity (EC) were conducted to investigate physical- 98 chemical and mechanical properties of CCR-Portland cement-stabilized dredged sludge 99 (CPDS). Besides, an array of microstructure tests were conducted to fully reveal the pos- 100 sible curing mechanism. Furthermore, the carbon footprints of CPDS were calculated and 101 compared with those of cemented dredged sludge. Finally, the solidification mechanism 102 of CPDS is discussed and revealed in this research."

3. Explain more about the variation of "The dredged sludge was collected in a construction site in Dawang High-tech Zone, 106 Zhaoqing City, Guangdong Province, China (Figure 1)."

4. How to control the mixes during mixing "he mixture design of stabilized dredged sludge is described in Table 3 based on 137 engineering knowledge of dry jet mixing applications."?

5. Recheck "Besides, according to the UCS 218 of CPDS cure for 28 d, the role of CCR for them in the strength development of CPDS is 219 observably different. The UCS of CPDS with 20% cement content initially decreases and 220 then increases as the CCR content increases from 0% to 10%, while the UCS with 10% and 221 30% cement content have sustainable growth."

6. Explain more "Furthermore, if the Ca(OH)2 concentration is excessive, the thickening of 324 alkali solution will reduce the ion mobility from the soil particles, preventing further ion 325 leaching [38]. Thus, it may be concluded that too less or too much alkali solution inhibits 326 the hydrolysis and precipitation of Si4+ and Al3+ ions from soil particles, as well as the 327 dissolution of Ca(OH)2 [39], further preventing the volcanic ash reaction and the develop- 328 ment in strength."

Reviewer 3 Report

The manuscript “Effect of calcium carbide residue on strength development along with mechanisms of cement-stabilized dredged sludge” studied the use of calcium carbide residue, as a solid alkaline activator on the strength development in CCR-Portland cement-stabilized dredged sludge. Τhe strength development of CPDS were investigated using different mechanical, physical and petrographic methods (UCS, pH, EC, XRD, SEM). The authors have employed correctly the techniques. I have found the methodological approach correct and the results are properly interpreted. The presentation of the problem is clear, the results correctly presented and the conclusions well explained. This manuscript is well written and well structured, with a proper English grammar and syntax. From a scientific point of view the manuscript shows a high degree of soundness, supported by pertinent references. To conclude, I suggest this manuscript to be published in the journal “Materials” after the below minor revisions:

Ø  The first section introduces the problem to a reader. It is well written, concise and informative. Some references should be added in the introduction for the use of different by-products using as cement additives. References to be cited in the introduction field:

·       Valorization of Slags Produced by Smelting of Metallurgical Dusts and Lateritic Ore Fines in Manufacturing of Slag Cements. Applied Sciences 2020, 10(13), 4670.

·       Evaluation of Cement Performance Using Industrial Byproducts Such as Nano MgO and Fly Ash from Greece. Applied Sciences 2021, 11(24), 11601.

Ø  Please provide the manufacturers for all the equipment used.

Ø  It would be convenient to include a table of abbreviations.

Round 2

Reviewer 2 Report

Accepted.